# Design and Simulation Analysis of a Reverse Flexible Harvesting Device for Fresh Corn

**Hongmei Zhang, Bo Chen, Zhijie Li, Chenhui Zhu, E Jin and Zhe Qu \***

College of Mechanical and Electrical Engineering, Henan Agricultural University, Zhengzhou 450002, China
* Correspondence: quzhe071171@henau.edu.cn

**Abstract:** Aiming at the problem of grain breakage during the harvesting of fresh corn, this paper theoretically analyzes the collision process between the ear picking device and the corn ear, and a flexible ear picking structure composed of flexible materials and buffer springs is determined. Combined with a new harvesting method that reverses the growth direction from top to bottom, a reverse flexible ear plucking device for fresh corn was designed. We used the ADAMS software to simulate the ear picking process of fresh corn, analyze the contact force between the rigid structure and flexible buffer structure under different picking claw speeds and stalk feeding speeds, and obtain the optimal parameter combination: the picking claw speed was 2 m/s, and the stalk feeding speed was 1 m/s. On the basis of the simulation, a reverse flexible fresh corn harvesting bench was built, and the optimal operating parameters were obtained from the test: the speed of ear picking claws was 2.11 m/s; the number of ear picking claws was four; the thickness of the flexible body was 4.52 mm; the stem feeding speed was 1.04 m/s; the corresponding grain breakage rate was 0.128%, which was far lower than the national standard (0.5%); and the ear impurity content was 0.3%, which was far lower than the national standard (2%). The results are consistent with the simulation results, proving that the model is reliable. This research achieved the harvest of fresh corn ears with a low grain damage rate, verified the possibility of reverse flexible ear picking, and provided a reference for the research and development of low-damage fresh corn harvesting machines.

**Keywords:** fresh corn; reverse direction; flexible ear picking; ADAMS simulation

## 1. Introduction

The corn ear picking device is the core of the corn harvester [1]. At present, there are three types of devices that can be used for fresh corn harvesting in China. First, traditional corn harvesting devices, which are mostly the ear picking roller type [2–4], are simple in structure, reliable in operation, and good in stability, but they can bite corn ears significantly, resulting in a high grain damage rate at the bottom of the ear, and easily cause secondary damage. The second is the improved conventional harvesting device. Scholars, such as Geng [5], Shang [6], and Wang [7], optimized the roller ear picking device, which can reduce the damage of corn kernels during harvest to a certain extent. Li [8] designed a combined corn ear picking mechanism with adjustable clearance to reduce the impurity content. Li [9] designed an annular ear picking device. The use of flexible clamping reduced the corn grain crushing rate, but the impurity content was high. Cheng [10] and Xu [11] established a pull-up corn harvesting test bench, which avoided fruit gnawing and reduced the rate of grain breakage, but the stem was easily pinched and broken. The third is the bionic ear picking device. Zhang [12] and Chen [13] proposed a bionic ear picking hand type of corn harvesting device, which can effectively reduce the rate of grain breakage, but the ear impurity rate is high. Zhu et al. [14,15] designed a bionic bending ear picking device, which can reduce the rate of grain damage, ear impurity content, and power consumption. However, it is easy to miss picking when harvesting double-eared plants.

In other parts of world, research has also been conducted to address these problems. Maria Kondoyanni et al. [16] conducted research on bionic harvesting, proposing a soft

robot gripper that imitates the human hand and makes it possible to mitigate the risk of surface bruises, rupture, the crushing destruction of plant tissue, and plastic deformation when harvesting soft-skinned fruits. Donato Romano et al. [17] also studied bionic robots to improve the control performance of bionic robot systems.

Nevertheless, the existing corn harvesting technology cannot meet the requirements of low ear damage for fresh corn harvesting well, and it is of great significance to solve the problems of high grain breakage rate [18–21]; the problem of high impurity content also needs to be solved. In this paper, a new type of harvesting method with reverse growth direction from top to bottom is adopted, combined with a new flexible structure composed of flexible materials and buffer springs. Thus, a reverse flexible ear plucking device for fresh corn is designed. The process of fresh corn picking was simulated by ADAMS software, and a virtual orthogonal test was arranged to analyze the contact force between the ear picking claws and the ear under rigid and flexible structures. On the basis of the simulation, a reverse flexible fresh corn harvesting bench was built, and the optimal operating parameters were obtained through experiments, in order to provide a reference for innovative research on corn harvesting with low damage and low impurity content.

## 2. Force Analysis of Panicle Collision Process

When the ear picking device collides with the corn ear, a considerable impact force appears instantly. When the external force impulse is constant, the peak impact force and impact time $\Delta t$ has a lot to do with it. The extension of the collision time can significantly reduce the impact force at the moment of collision, thereby reducing the damage to the grains. Considering the corn ear as a particle, its collision process conforms to the impulse-momentum theorem:

$$I = \int_{t_0}^{t_0+\Delta t} ma(t)dt \tag{1}$$

In the formula, $I$ is the impact on corn ears, N·s; $m$ is the mass of corn ears, kg; $a$ is the acceleration, m/s$^2$; and $t$ is the time, s.

Note $y$ is the vertical coordinate of the middle of the fruit ear. During the collision, the motion equation of this single-degree-of-freedom system during contact is:

$$\begin{aligned} m\ddot{y} + c\dot{y} + (k + k_o)y = 0 \\ y(0) = 0, \dot{y}(0) = v_0 \end{aligned} \tag{2}$$

In the case of underdamping, the solution of Equation (2) is:

$$y = Ae^{-\lambda\omega_i t}\sin\omega_s t \tag{3}$$

where

$$\lambda = \frac{c}{2m\omega_i} \tag{4}$$

$$A = \frac{v_0}{\omega_s} \tag{5}$$

$$\omega_s = \sqrt{1 - \lambda^2}\omega_i \tag{6}$$

$$\omega_i = \sqrt{\frac{k + k_0}{m}} \tag{7}$$

In the formula, $\lambda$ is the viscous damping ratio, $\lambda < 1$; $A$ is the amplitude, mm; $\omega_s$ is the damped natural frequency, Hz; and $\omega_i$ is the undamped natural frequency, Hz.

With an initial displacement of 0, the collision time is determined by the smallest positive root of Equation (8):

$$f(t) = c\dot{y} + ky = 0 \tag{8}$$

The available collision time is:

$$\Delta t = \frac{\pi + \arctan\lambda_2}{\omega_i\sqrt{1 - \lambda^2}}$$ (9)

where

$$\lambda_2 = \frac{2\lambda\sqrt{1 - \lambda^2}}{k_0/\omega_i^2 - (1 - 2\lambda^2)}$$ (10)

It can be seen from Equations (9) and (10) that the impact time is related to the mass, stiffness coefficient, damping coefficient, and other parameters of the system, and these inherent parameters depend on its structure and materials. Therefore, by changing the structure and contact materials of the collision system, the stiffness coefficient of the system is reduced, and the damping coefficient is increased, thereby prolonging the collision contact time and reducing the rate of grain damage.

## 3. Structure and Working Principle of Reverse Flexible Fresh Corn Harvesting Device

Following the design principles of reducing the stiffness coefficient and increasing the damping coefficient, this paper designed a reverse flexible ear picking device for fresh corn with flexible surfaces and buffer springs as the flexible structures and the harvest direction of reverse growth from top to bottom. The reverse flexible harvesting device for fresh corn consists of a motor, internal and external racks, a stalk pulling device, flexible ear picking claws, and a grain holding device, among which the flexible ear picking claws are evenly and symmetrically distributed on the chains on both sides. The reverse flexible fresh corn ear picking device is shown in Figure 1.

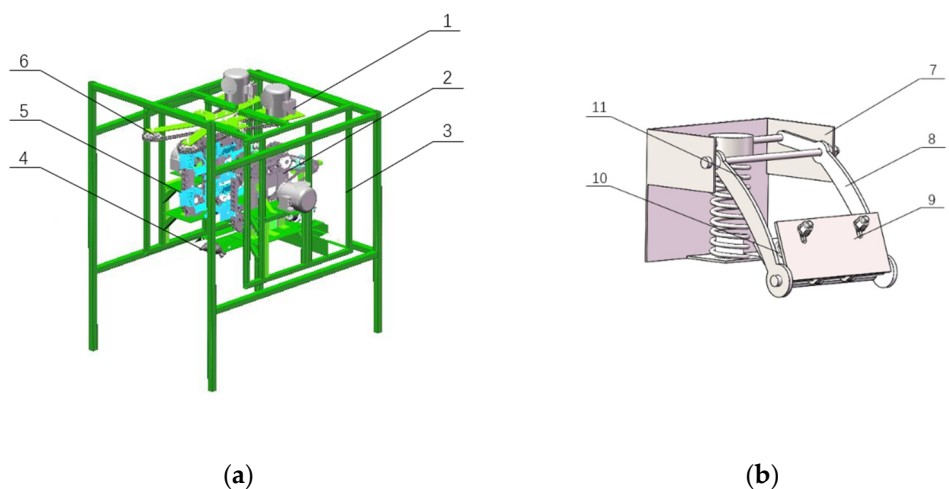

(**a**)  (**b**)

**Figure 1.** Reverse flexible earing device for fresh corn. (**a**) Overall structure of ear picking device; (**b**) flexible ear picking claw structure. 1. Motor. 2. Inner frame. 3. Outer frame. 4. Stem pulling device. 5. Flexible ear picking claw. 6. Reel holding device. 7. Ear picking frame. 8. Support plate. 9. Up and down ear picking plates. 10. Buffer spring. 11. Buffer spring.

When working, the corn plants are fed by the feeding device, and the upper weeding device and the picking roll feeding section in lower stalk pulling device work together to ensure that the fresh corn plants are upright. At the same time, driven by the chain, the flexible ear picking claws move from top to bottom, colliding with the corn ear, and the ear is subjected to downward force, which breaks off the plant at the connection point between the fruit stalk and the stem to complete the ear picking process.

Due to the buffering effect of the flexible surface and spring, the collision energy is converted into the potential energy of the spring and flexible body and dissipated, reducing the grain breakage of the ear. The corn stalk moves backward to the ear picking section of the ear picking roller, which can be used for secondary ear picking of the missed ear.

## 4. Simulation of the Picking Process of Fresh Corn

### 4.1. Construction of the Simulation Model

The simulation model of reverse flexible ear picking device and corn plant were established by using ADAMS software. Since any ear picking claw only acts on one corn in one ear picking stroke, the ear picking process was simplified, and the number of ear picking claws was simplified to one. In order to improve the quality of discrete element simulation, it was necessary to select reasonable material contact model and correct boundary parameters. Solid-to-solid was selected as the contact type between corn ears and ear picking claws, and impact was selected as the contact force type. Due to limited references on the contact parameters between corn straw and other materials, wood with similar physical properties to straw was used instead. The specific contact parameters [22,23] are shown in Table 1.

**Table 1.** Contact parameters of the different materials.

| Material Type | Stiffness /(N·mm$^{-1}$) | Force Exponent | Damping /(N·s·mm$^{-1}$) | Penetration Depth/(mm) | Static Coefficient | Dynamic Coefficient |
|---|---|---|---|---|---|---|
| Steel–wood | 2855 | 1.5 | 0.57 | 0.1 | 0.3 | 0.25 |
| Rubber–wood | 2855 | 1.1 | 0.57 | 0.1 | 0.5 | 0.25 |

We conducted the following procedure: Apply different bending forces to the fresh corn ear and analyze the change in connection force between the components of the fresh corn plant. It can be obtained that when the Y-axis displacement change value of Bushing1, the flexible connection force between the corn stalk and the fruit stalk, reaches 2.3106 mm, the corn ear will fall. Use sensor1 to monitor the real-time data of Bushing1's displacement on the Y-axis. When the measured value was greater than 2.3106 mm, it was determined that an event occurs, and the sensor1 and Bushing1 are set to fail at this time. The ear handle bends and breaks at the connection with the straw. The corn ear falls with the ear handle breaking, and the heading process is completed. Figure 2 shows the simulation process of picking ears.

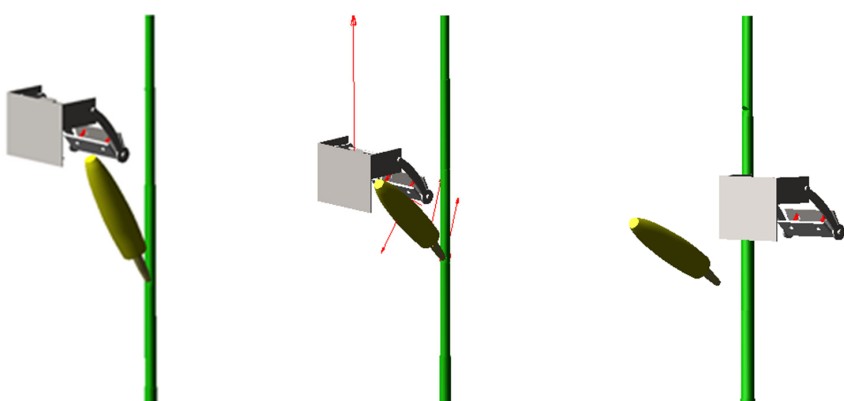

**Figure 2.** Simulation process of picking ears.

### 4.2. Design of the Simulation Test

The quadratic regression orthogonal rotation center combination test method was used to carry out simulation optimization test research, and the forces of flexible buffer mechanism and rigid mechanism (flexible buffer mechanism removes flexible body and spring) on fresh corn ears under different picking claw speed and straw feeding speed were compared to obtain the combination of optimal parameters. The coding table of the experimental factors is shown in Table 2.

Figures 3–7 show the real-time change curves of the impact force on the corn ear during the contact between the flexible ear picking claws and the fresh corn plants. Each group of data

is composed of two parts: the left side is the contact force change curve of the rigid mechanism, and the right side is the contact force change curve of the flexible buffer mechanism.

**Table 2.** Codes of the experimental factors.

| Codes | Factors | |
|---|---|---|
| | Corn Plant Feeding Speed/(m·s$^{-1}$) | Ear Picking Claw Speed/(m·s$^{-1}$) |
| 1.147 | 0.29 | 1.29 |
| 1 | 0.5 | 1.5 |
| 0 | 1 | 2 |
| −1 | 1.5 | 2.5 |
| −1.147 | 1.71 | 2.71 |

The comparative analysis of test data shows that:

(1) By comparing the left and right simulation test data, it was found that the overall curve change law of the two mechanisms is similar, but compared with the rigid mechanism, the flexible buffer mechanism has an obvious effect on reducing the impact force by prolonging the action time.

(2) Comparing five groups of different claw picking speeds, it was found that, when the picking speed of the claw is 1.5 m/s, it generates the minimum collision force (Figures 3a and 5a), followed by 2 m/s and 2.5 m/s, and the maximum collision force is 2.71 m/s. It can be seen that the picking speed of the claw is the key factor to determine the size of the impact force, but the size of the impact force is not completely proportional to the picking speed of the claw. The best ear picking effect can be achieved only when the feeding speed and the claw picking speed achieve a certain match.

(3) The five groups of different feeding speeds were compared, and it was found that, when the feeding speed is 0.5 m/s, the smallest impact force is produced (Figure 3a), followed by 1 m/s, and the maximum impact force is achieved when the speed reaches 1.5 m/s. This shows that the feeding speed is an important factor affecting the magnitude of the collision force.

To summarize, and taking into account the ear picking efficiency and collision damage, the ear picking effect was the best when the claw picking speed was 2 m/s and the corn feeding speed was 1 m/s.

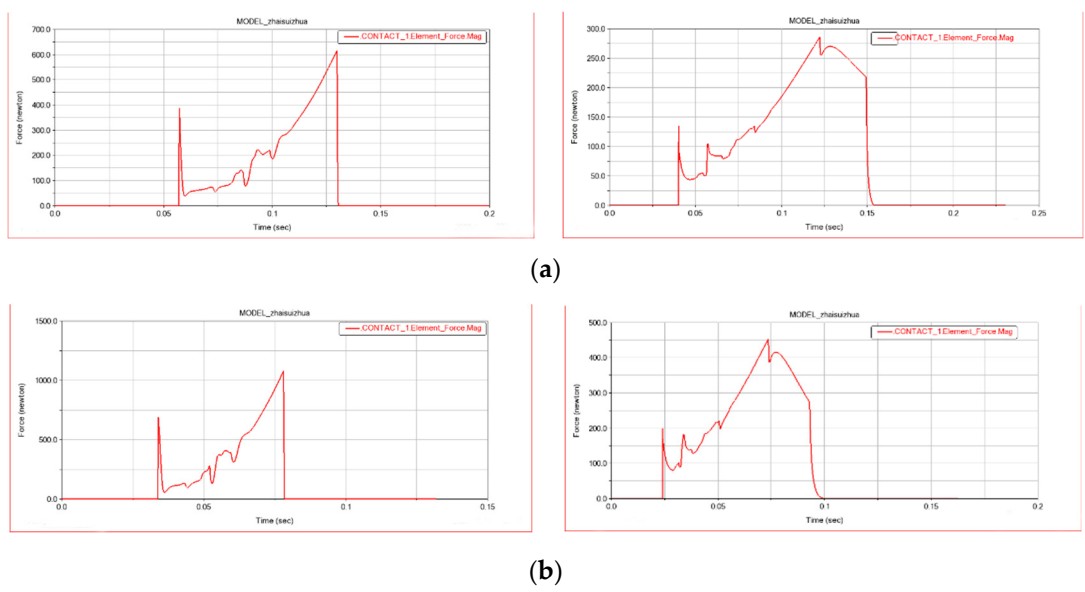

**Figure 3.** Contact force of ears at different claw picking speeds with a forward speed of 0.5 m/s. (**a**) Claw picking speed: 1.5 m/s; (**b**) claw picking speed: 2.5 m/s.

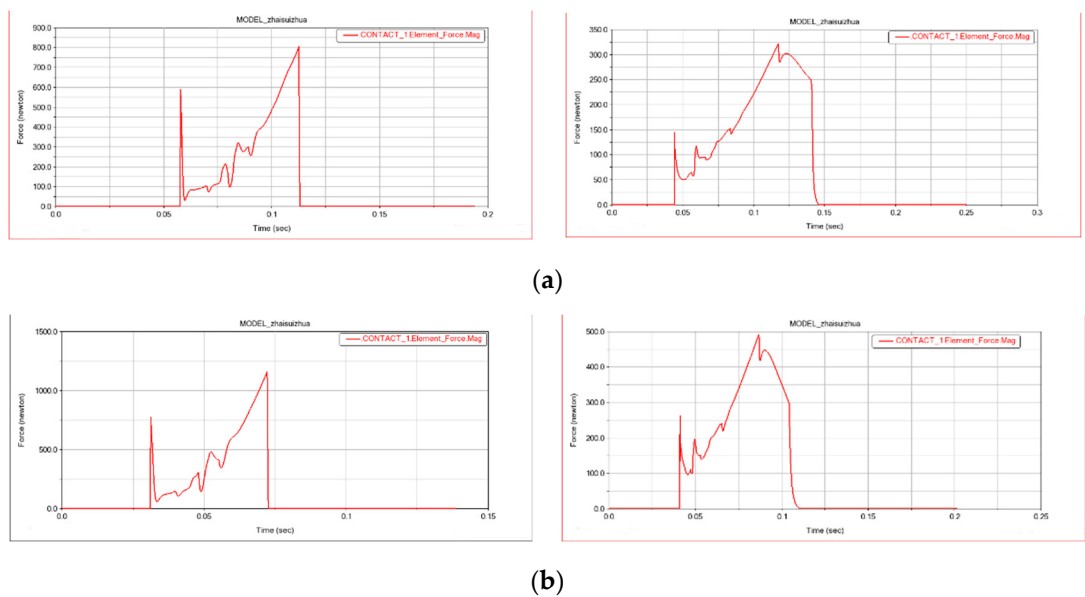

**Figure 4.** Contact force of ears at different claw picking speeds with a forward speed of 1 m/s. (**a**) Claw picking speed: 2 m/s; (**b**) claw picking speed: 2.71 m/s.

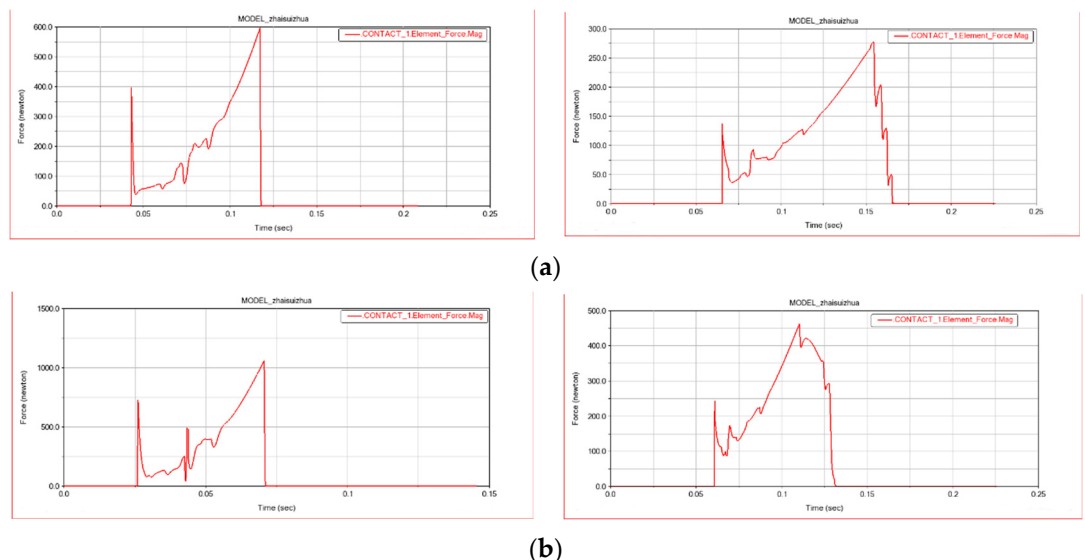

**Figure 5.** Contact force of ears at different claw picking speeds with a forward speed of 1.5 m/s. (**a**) Claw picking speed: 1.5 m/s; (**b**) claw picking speed: 2.5 m/s.

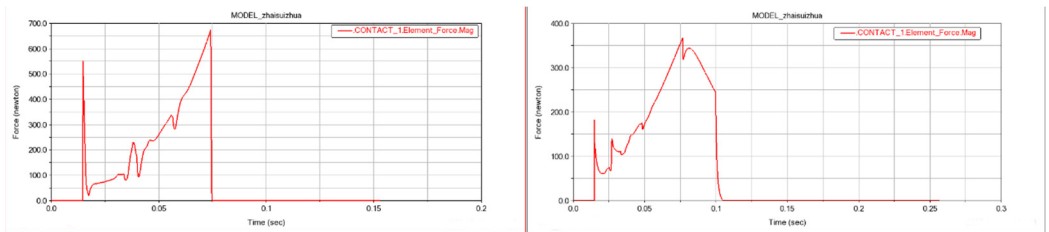

**Figure 6.** Contact force on the ear when the forward speed is 0.29 m/s, and the claw picking claw speed is 2 m/s.

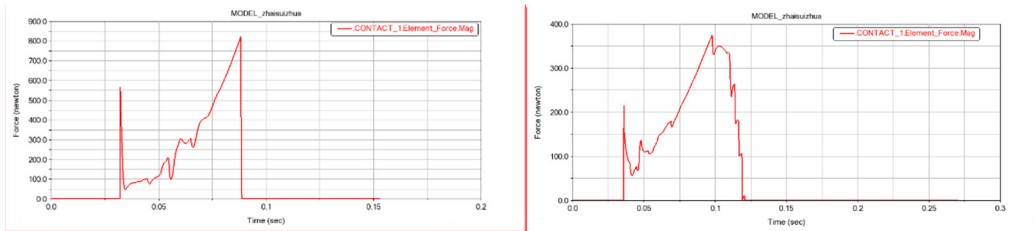

**Figure 7.** Contact force on the ear when the forward speed is 1.71 m/s and the claw picking speed is 2 m/s.

## 5. Optimization Experiment of the Ear Picking Effect

### 5.1. Test Conditions and Equipment

The experiment selected the whole mature colored sweet glutinous fresh corn plant planted in Xinxiang, Henan Province. The instruments and equipment included the processed reverse flexible fresh corn ear picking bench, computer, digital tachometer (TM690 contact type), variable speed frequency modulation (Delixi CDI-EM60/61 series), high-speed camera system (CamRecord 1000), and three-phase motor (YCT112-4 A type). The reverse flexible test bench for harvesting fresh corn is shown in Figure 8.

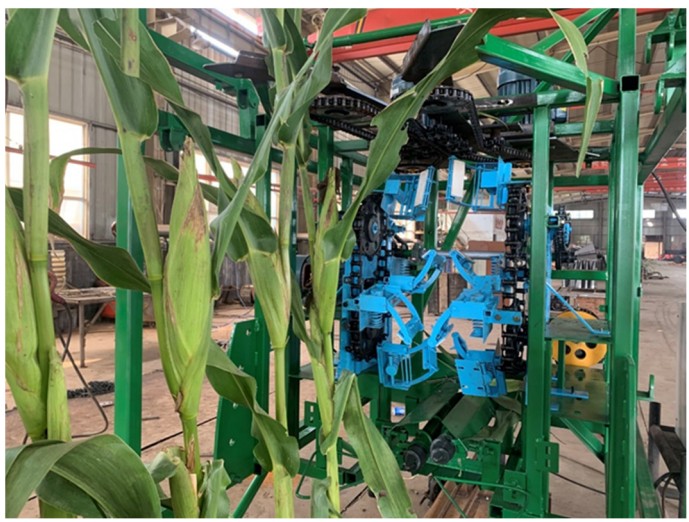

**Figure 8.** Ear picking test bench for fresh corn.

During the test, the reverse flexible fresh corn picking test bench was fixed, and the corn plants were continuously fed through the feeding device to simulate the relative movement of the harvester and the corn in actual field operations. When the corn plant enters the ear picking area, the flexible ear picking claws are driven by the chain and moved from top to bottom to pick the corn ear. When the next corn plant enters the ear picking area, it is picked by the next flexible ear picking claw.

### 5.2. Bench Test

#### 5.2.1. Experimental Method

According to the structure, the working parameters, and previous theoretical calculation of the reverse flexible fresh corn harvesting bench, the grain breakage rate and ear impurity ratio were selected as experimental indicators in this experiment, and four key influencing factors were selected; their level ranges were as follows: speed of ear picking claws is 1.5–2.5 m/s, number of ear picking claws was 3–5, flexible body thickness was 2–6 mm, and stalk feeding speed was 0.5–1.5 m/s.

5.2.2. Experimental Design

Four factors and three levels of the Box–Behnken response surface analysis methods [24–27] were used in the test. The test factors, such as the speed of the picking claw, the number of picking wheels, the thickness of flexible body, and the speed of straw feeding, were expressed as A, B, C, and D, respectively, and the test indicators, such as the grain crushing rate and the impurity content rate of the ears, were expressed as $W_1$ and $W_2$, respectively. The coding table of test factors is shown in Table 3. A total of 29 groups were created, with three experiments for each group, each with five corn plants, and an average of the results of the three experiments was made. The experimental program was designed, and the results were analyzed using the Design-Expert 8.0.6 software. Table 4 shows the experimental results.

**Table 3.** Codes of the experimental factors.

| Codes | Factors | | | |
| --- | --- | --- | --- | --- |
| | Ear Picking Claw Speed A/(m·s$^{-1}$) | Number of Claws B | Flexible Body Thickness C | Corn Plant Feed Speed D/(m·s$^{-1}$) |
| 1 | 1.5 | 3 | 2 | 0.5 |
| 0 | 2 | 4 | 4 | 1 |
| −1 | 2.5 | 5 | 6 | 1.5 |

**Table 4.** Program and results of the test of the quadratic rotation–orthogonal combination.

| No. | Factors | | | | Grain Breakage Rate $W_1$/(%) | Ear Impurity Content $W_2$/(%) |
| --- | --- | --- | --- | --- | --- | --- |
| | Ear Picking Claw Speed A/(m·s$^{-1}$) | Number of Claws B | Flexible Body Thickness C | Corn Plant Feed Speed D/(m·s$^{-1}$) | | |
| 1 | −1 | 0 | 0 | −1 | 0.14 | 0.43 |
| 2 | 0 | 1 | 0 | −1 | 0.17 | 0.59 |
| 3 | −1 | 0 | −1 | 0 | 0.2 | 0.31 |
| 4 | 0 | −1 | −1 | 0 | 0.19 | 0.68 |
| 5 | −1 | 0 | 1 | 0 | 0.32 | 0.39 |
| 6 | 1 | 0 | 0 | −1 | 0.21 | 0.51 |
| 7 | 1 | 1 | 0 | 0 | 0.25 | 0.62 |
| 8 | 0 | 0 | −1 | −1 | 0.16 | 0.41 |
| 9 | 0 | −1 | 0 | −1 | 0.13 | 0.78 |
| 10 | 0 | 0 | 0 | 0 | 0.11 | 0.27 |
| 11 | 0 | 0 | −1 | 1 | 0.16 | 0.46 |
| 12 | 0 | −1 | 1 | 0 | 0.27 | 0.7 |
| 13 | 1 | 0 | 1 | 0 | 0.33 | 0.38 |
| 14 | 1 | 0 | 0 | 1 | 0.23 | 0.5 |
| 15 | 1 | 0 | −1 | 0 | 0.3 | 0.38 |
| 16 | 1 | −1 | 0 | 0 | 0.28 | 0.76 |
| 17 | 0 | 0 | 1 | −1 | 0.2 | 0.45 |
| 18 | −1 | 1 | 0 | 0 | 0.24 | 0.5 |
| 19 | 0 | 0 | 0 | 0 | 0.12 | 0.29 |
| 20 | 0 | 1 | −1 | 0 | 0.24 | 0.5 |
| 21 | 0 | 1 | 0 | 1 | 0.18 | 0.67 |
| 22 | 0 | 0 | 0 | 0 | 0.14 | 0.3 |
| 23 | 0 | 0 | 0 | 0 | 0.13 | 0.31 |
| 24 | 0 | 1 | 1 | 0 | 0.26 | 0.54 |
| 25 | 0 | −1 | 0 | 1 | 0.15 | 0.79 |
| 26 | −1 | −1 | 0 | 0 | 0.19 | 0.73 |
| 27 | 0 | 0 | 1 | 1 | 0.26 | 0.48 |

5.3. *Analysis of the Test Results*

5.3.1. Regression Analysis

(1)　Establishing the regression model of grain breakage rate

Using Design-Expert 8.0.6 software to regress and fit the test results, we obtained the regression mathematical model of grain breaking rate $W_1$:

$$W_1 = 1.964 - 1.059A - 0.208B - 0.118C - 0.012D - 0.04AB - 0.023AC - 0.0075BC$$
$$+0.015CD + 0.314A^2 + 0.038B^2 + 0.02C^2 - 0.036D^2 \tag{11}$$

The coefficient of determination $R^2$ of the regression equation was 0.984, indicating a high degree of fitting; the regression model was analyzed by variance, and the results are shown in Table 5. It can be observed that $p < 0.0001$ of the model, indicating that the regression equation is significant and can describe the relationship between each factor and response value; the lack of fit was $p = 0.663 > 0.05$, showing that the residual item is not significant, and there are no other main factors affecting the results, so the regression model was established. $p < 0.01$ was set for A, B, C, D, AB, AC, $A^2$, $B^2$, and $C^2$, which has a very significant impact on the results; $p < 0.05$ was for the number of ear picking claws, BC, CD, and $D^2$, indicating that it has a significant impact on the results; other $p > 0.05$ was for factors and interaction terms, which had no significant effect on the results. The order of significance of each factor to the screening efficiency from large to small was the claw picking speed, the thickness of the flexible body, the stalk feeding speed, and the number of picking claws. According to the regression equation, the influence of interaction of various factors on the results is shown in Figure 9.

When the number of ear picking claws and the stalk feeding speed were constant, the effect of the interaction between the ear picking claws speed and the thickness of the flexible body on the grain breakage rate is shown in Figure 9a. When the ear picking claw speed is constant, the thickness of the flexible body is too large or too small, which will lead to an increase in the rate of grain breakage. When the thickness of flexible body is constant, the rate of grain breakage increases with the increase in ear picking claw speed.

When the thickness of the flexible body and the feeding speed of the stalk are constant, we measured the effect of the interaction between the speed of the picking claws and the number of claws on the grain breakage rate, shown in Figure 9b. When the number of ear picking claws is constant, with the increase in ear picking claws, the rate of grain breakage will gradually increase. When the ear picking claw speed was constant, the grain breakage rate increased slightly with the decrease in the number of ear picking claws, and the change was not obvious.

**Table 5.** Variance analysis of grain breakage rate.

| Source | Sum of Squares | df | Mean Squares | F Value | *p*-Value Prob > F |
|---|---|---|---|---|---|
| Model | 0.11 | 12 | $9.178 \times 10^{-3}$ | 84.15 | <0.0001 ** |
| A | $9.075 \times 10^{-3}$ | 1 | $9.075 \times 10^{-3}$ | 83.21 | <0.0001 ** |
| B | $1.408 \times 10^{-3}$ | 1 | $1.408 \times 10^{-3}$ | 12.91 | 0.0024 ** |
| C | 0.013 | 1 | 0.013 | 116.22 | <0.0001 ** |
| D | $1.875 \times 10^{-3}$ | 1 | $1.875 \times 10^{-3}$ | 17.19 | 0.0008 ** |
| AB | $1.600 \times 10^{-3}$ | 1 | $1.600 \times 10^{-3}$ | 14.67 | 0.0015 ** |
| AC | $2.025 \times 10^{-3}$ | 1 | $2.025 \times 10^{-3}$ | 18.57 | 0.0005 ** |
| BC | $9.000 \times 10^{-4}$ | 1 | $9.000 \times 10^{-4}$ | 8.25 | 0.0111 * |
| CD | $9.000 \times 10^{-4}$ | 1 | $9.000 \times 10^{-4}$ | 8.25 | 0.0111 * |
| $A^2$ | 0.040 | 1 | 0.040 | 365.72 | <0.0001 ** |
| $B^2$ | $9.573 \times 10^{-3}$ | 1 | $9.573 \times 10^{-3}$ | 87.78 | <0.0001 ** |
| $C^2$ | 0.042 | 1 | 0.042 | 389.41 | <0.0001 ** |
| $D^2$ | $5.352 \times 10^{-4}$ | 1 | $5.352 \times 10^{-4}$ | 4.91 | 0.0416 |
| Residual | $1.745 \times 10^{-3}$ | 16 | $1.091 \times 10^{-4}$ | | |
| Lack of Fit | $1.225 \times 10^{-3}$ | 12 | $1.021 \times 10^{-4}$ | 0.79 | 0.6663 |
| Pure Error | $5.200 \times 10^{-4}$ | 4 | $1.300 \times 10^{-4}$ | | |
| Cor Total | 0.11 | 28 | | | |

Note: $p < 0.01$ (extremely significant, **); $p < 0.05$ (significant, *).

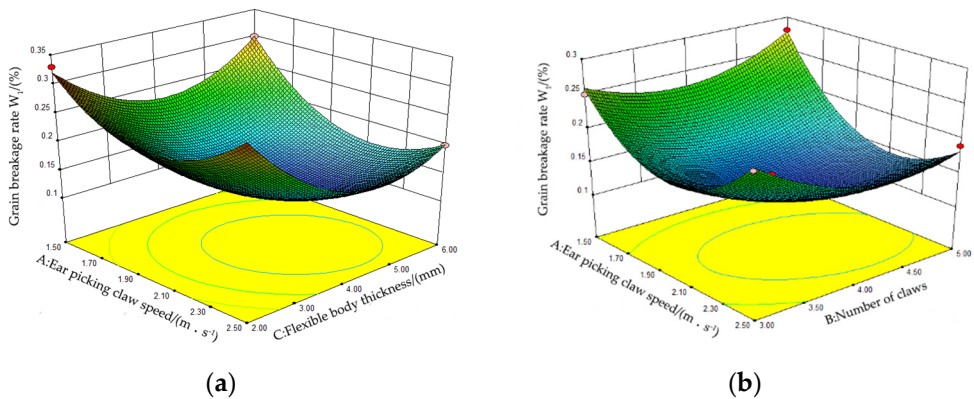

**Figure 9.** Effects of the interaction of factors on the grain breaking rate. (**a**) Interaction between ear picking claw speed and flexible body thickness; (**b**) interaction between ear picking claw speed and number of ear picking claws.

(2)　Establish a regression model of ear impurity rate

　　Using the Design-Expert 8.0.6 software to perform regression fitting processing on the test results, we obtained the regression mathematical model of the ear trash content $W_2$:

$$W_2 = 6.559 - 1.046A - 2.379B - 0.008C - 1.067D + 0.045AB - 0.02AC - 0.08AD \\ +0.035BD + 0.244A^2 + 0.292B^2 + 0.0049C^2 + 0.524D^2 \tag{12}$$

The determination coefficient $R^2$ of the regression equation is 0.995, with a high degree of fitting. The results of variance analysis of the regression model are shown in Table 6. It can be observed that $p < 0.0001$ of the model indicates that the regression equation is significant; with a misfit item of $p = 0.5180 > 0.05$, the residual item is not significant, and no other major factors affect the results, so the regression model is established. $p < 0.01$ is for the quadratic terms of A, B, C, D, AB, $A^2$, $B^2$, $C^2$, and $D^2$, which has a very significant impact on the results; $p < 0.05$ is for AC, AD, and BD, indicating that it has a significant impact on the results; other factors and $p > 0.05$ were for the interaction term, which had no significant effect on the results. The order of significance of each factor on ear impurity content from large to small was the number of ear picking claws, the speed of ear picking claws, the feeding speed of stalk, and the thickness of the flexible body. According to the regression equation, the influence of the interaction of various factors on the results is shown in Figure 10.

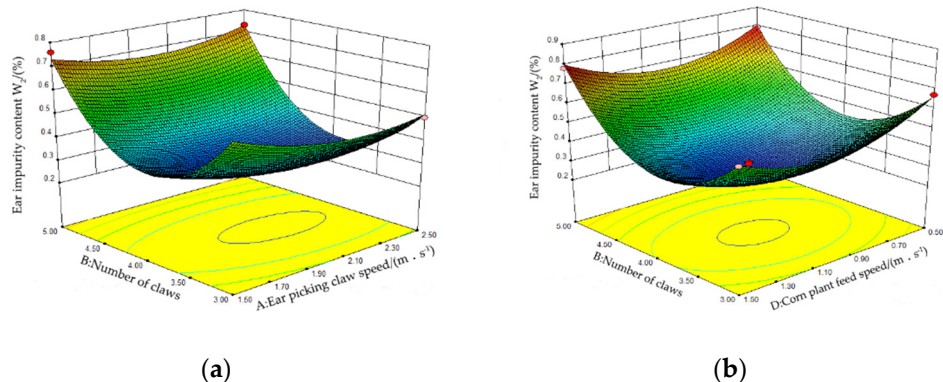

**Figure 10.** Effect of the interaction of factors on the impurity content of the ears. (**a**) Interaction between the speed and number of picking claws; (**b**) interaction between the straw feeding speed and number of picking claws.

**Table 6.** Variance analysis of the ear impurity content.

| Source | Sum of Squares | df | Mean Squares | F Value | *p*-Value Prob > F |
|---|---|---|---|---|---|
| Model | 0.72 | 12 | 0.060 | 84.15 | <0.0001 ** |
| A | $7.008 \times 10^{-3}$ | 1 | $7.008 \times 10^{-3}$ | 83.21 | <0.0001 ** |
| B | 0.087 | 1 | 0.087 | 12.91 | 0.0024 ** |
| C | $3.333 \times 10^{-3}$ | 1 | $3.333 \times 10^{-3}$ | 116.22 | <0.0001 ** |
| D | $4.408 \times 10^{-3}$ | 1 | $4.408 \times 10^{-3}$ | 17.19 | 0.0008 ** |
| AB | $2.025 \times 10^{-3}$ | 1 | $2.025 \times 10^{-3}$ | 14.67 | 0.0015 ** |
| AC | $1.600 \times 10^{-3}$ | 1 | $1.600 \times 10^{-3}$ | 18.57 | 0.0005 ** |
| AD | $1.600 \times 10^{-3}$ | 1 | $1.600 \times 10^{-3}$ | 8.25 | 0.0111 * |
| BD | $1.225 \times 10^{-3}$ | 1 | $1.225 \times 10^{-3}$ | 8.25 | 0.0111 * |
| $A^2$ | 0.024 | 1 | 0.024 | 365.72 | <0.0001 ** |
| $B^2$ | 0.55 | 1 | 0.55 | 87.78 | <0.0001 ** |
| $C^2$ | $2.552 \times 10^{-3}$ | 1 | $2.552 \times 10^{-3}$ | 389.41 | <0.0001 ** |
| $D^2$ | 0.11 | 1 | 0.11 | 4.91 | 0.0416 * |
| Residual | $3.738 \times 10^{-3}$ | 16 | $2.336 \times 10^{-4}$ | | |
| Lack of Fit | $2.858 \times 10^{-3}$ | 12 | $2.382 \times 10^{-4}$ | 1.08 | 0.5180 |
| Pure Error | $8.800 \times 10^{-4}$ | 4 | $2.200 \times 10^{-4}$ | | |
| Cor Total | 0.72 | 28 | | | |

Note: $p < 0.01$ (extremely significant, **); $p < 0.05$ (significant, *).

When the stalk feeding speed and the thickness of the flexible body are constant, the effect of the interaction between the speed of the picking claws and the number of ear picking claws is shown in Figure 10a. When the ear picking claws were fixed at a certain speed, with the increase in the number of ear picking claws, the ear impurity content would gradually increase. When the number of ear picking claws was constant, the ear impurity content decreased slightly with the increase in ear picking claws, and the change was not obvious.

When the thickness of the flexible body and the speed of picking claws are fixed, we measured the effect of the interaction between the feeding speed of the stem and the number of claws on the ear, shown in Figure 10b. When the number of ear picking claws was constant, the rate of grain breakage decreased gradually with the increase in stalk feeding rate. When the stalk feeding speed was constant, the ear impurity content increased significantly with the increase in the number of ear picking claws.

### 5.3.2. Parameter Optimization

In order to achieve better harvest effect, the optimization module was used to optimize the regression model [28,29] and set the solution target as the minimum response value, and the optimal parameter combination was obtained as follows: harvest claw speed of 2.11 m/s; number of harvest claws, four; flexible body thickness of 4.52 mm; stalk feeding speed of 1.04 m/s, with a corresponding mean grain crushing rate of 0.12%; and a mean ear impurity content of 0.29%, which were the best test indicators.

### 5.4. Bench Test Verification

The optimized parameter combination was tested and verified, and the speed of the picking claw was 2.11 m/s, the number of picking claws was four, the thickness of flexible body was 4.52 mm, and the straw feeding speed was 1.04 m/s. Taking the average value of 10 trials, and the grain loss rate was 0.128%, and the relative error of the predicted value was 0.008%, which is far lower than the national standard of 0.5% [30]. The average impurity content of the ear was 0.3%, and the relative error of the predicted value is 0.01%, which is far lower than the national standard of 2%, which verifies the reliability of the model.

## 6. Conclusions

(1) The collision process between the ear picking device and the corn ear was theoretically analyzed, and a reverse flexible fresh corn ear picking device was designed. The device effectively reduced the grain breakage rate and the ear impurity content in the harvest of fresh corn ears.

(2) Using ADAMS software to simulate the ear picking process of fresh corn, we analyzed the contact force between the rigid structure and flexible buffer structure under different claw picking speeds and stalk feeding speeds and obtained the optimal parameter combination: the picking claw speed was 2 m/s and the stalk feeding speed was 1 m/s.

(3) The reverse flexible fresh corn harvesting bench was built for experimental verification, and the experimental data were optimized and analyzed using Design-Expert 8.0.6. The optimal parameter combination of each experimental factor was obtained: the speed of picking claw was 2 m/s; the number of claws was four; the thickness of the flexible body was 4 mm; the feeding speed of the stem was 1 m/s, with a corresponding grain breakage rate of 0.12%; and the ear impurity content was 0.29%.

(4) The optimized parameter combination was selected, and the bench test was conducted again to verify that the grain crushing rate was 0.128%; the relative error of the predicted value was 0.008%, which is far lower than the national standard of 0.5%. The ear impurity content was 0.3%, and the relative error of the predicted value was 0.01%, which is far lower than the national standard of 2%. The optimized prediction model is reliable.

## 7. Patents

Zhu, C.H.; Chen B.; Zhang H.M.; Fan X.D.; Zhang C.M. A reverse flexible ear picking device for fresh corn: ZL202221291862.0 [P]. 23 September 2022.

**Author Contributions:** Conceptualization, H.Z. and Z.Q.; methodology, H.Z. and B.C.; investigation, Z.Q. and E.J.; Visualization, C.Z.; writing—original draft preparation, B.C. and Z.L. All authors have read and agreed to the published version of the manuscript.

**Funding:** This research was funded by Henan Province Modern Agricultural Industrial Technology System Maize Whole-process Mechanization Special Project (HARS-22-02-G4); National Key Research and Development Program (2018YFD0300704); and Henan Province Science and Technology Research (222102110457).

**Institutional Review Board Statement:** Not applicable.

**Data Availability Statement:** The data used to support the findings of this study are available from the corresponding author upon request.

**Acknowledgments:** The authors would like to thank their college and the laboratory, as well as gratefully appreciate the reviewers who provided helpful suggestions for this manuscript.

**Conflicts of Interest:** The authors declare no conflict of interest.

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
