# Peer review of "Design and Simulation Analysis of a Reverse Flexible Harvesting Device for Fresh Corn"

_agriculture, doi:10.3390/agriculture12111953_

Round 1

Reviewer 1 Report

The methodology shoudl be organized to be clearer, and the objective and originality of the study should be discussed.

Also, authors may discuss on the role of technology and robotics in agriculture (inlcuding crop breeding, aquafarming, animal rearing).

Many recent works focused on this, and also the role of biomimetic design may be discussed to reduce the stress to living organisms.

some examples 

Romano, D., Wahi, A., Miraglia, M., & Stefanini, C. (2022). Development of a Novel Underactuated Robotic Fish with Magnetic Transmission System. Machines10(9), 755.

Kondoyanni, M., Loukatos, D., Maraveas, C., Drosos, C., & Arvanitis, K. G. (2022). Bio-Inspired Robots and Structures toward Fostering the Modernization of Agriculture. Biomimetics7(2), 69.

Also, a deep english revision is needed.

Reviewer 2 Report

This paper studies and analyzes the collision process between the ear picking device and the corn ear. The Adams software was used to simulate the ear picking process of fresh corn, and the contact force between the rigid structure and the flexible buffer structure under different picking claw speeds and straw feeding speeds was analyzed, and the optimal parameter combination was obtained. Constructed reverse flexible fresh corn harvesting table. The optimal operating parameters are obtained through experiments. This paper provides a reference for the research and development of low-damage fresh corn harvester. While this works fine, there are some bugs in this article. Some details are shown below.

1. Introduction, "The existing corn harvesting technology cannot meet the requirements of low ear damage in fresh corn harvesting, which is of great significance to solving the problems of high ear gnawing and high breaking rate in fresh corn harvesting. Which specific thing can obviously solve the problem of fresh corn harvesting? There is no clear explanation for the high rate of corn ear breaking. Please provide additional explanations.

2. Section 3, Figure 1 illustrates the structure of the device. The working principle of the device is described in paragraph 2. However, the structure described in the previous paragraph does not correspond to the structure shown in FIG. 1 . Please correct.

3. The process of picking ears is shown in Figure 2. The contact process between the upper and lower ear pickup plates and the ear is missing. The process of picking ears is not fully shown. Please provide additional clarification.

4. What does "level" in Table 2 mean? Please provide additional clarification.

5. Figure 8 is a casting test rig for fresh corn. However, key equipment for the experiment is not specified in Figure 8. Please provide additional clarification.

6. Section 5.1 describes the test conditions and equipment. However, there is no specific description of the experimental procedure. Please provide additional clarification.

Round 2

Reviewer 1 Report

Authors addressed almost all my comments, and the manuscript is much improved.